# Characteristics and etiologies of hepatocellular carcinoma in patients without cirrhosis: When East meets West

Yi-Hao Yen[1]*, Yu-Fan Cheng[2], Jing-Houng Wang[1], Chih-Che Lin[3], Chih-Chi Wang[3]

**1** Division of Hepatogastroenterology, Department of Internal Medicine, Kaohsiung Chang Gung Memorial Hospital and Chang Gung University College of Medicine, Kaohsiung, Taiwan, **2** Liver Transplantation Center, Department of Diagnostic Radiology, Kaohsiung Chang Gung Memorial Hospital, Chang Gung University College of Medicine, Kaohsiung, Taiwan, **3** Liver Transplantation Center and Department of Surgery, Kaohsiung Chang Gung Memorial Hospital, Kaohsiung, Taiwan

\* cassellyen@yahoo.com.tw

## Abstract

**Data Availability Statement:** All relevant data are within the manuscript and its Supporting Information files.

### Background/Aims

A recent study from the United States reported that nearly 12% of hepatocellular carcinomas (HCCs) occurred in patients without cirrhosis.

Non-alcoholic fatty liver disease (NAFLD) was the most common liver disease in these patients. We aim to evaluate the characteristics, etiologies, and outcomes of cases of non-cirrhotic HCC in East Asia, where there is a higher prevalence of hepatitis B virus (HBV)-associated non-cirrhotic HCC.

### Methods

This retrospective study consecutively enrolled de novo HCC patients managed at our institution from 2011 to 2017. The presence of cirrhosis was assessed by histology; if histology was not available, it was assessed by image study.

### Results

2055 patients with HCC were enrolled in this study. Among them, 529 (25.7%) were non-cirrhotic. The non-cirrhotic patients were younger (60.9 vs. 62.5 years, p = 0.006), included a greater proportion of males (78.1% vs. 71.3%, p = 0.002), and had a lower body mass index (24.3 vs. 25.3 kg/m$^2$, p<0.001) than the cirrhotic patients. Among the non-cirrhotic patients, HBV was the most common liver disease (49.0%). The patients with non-cirrhotic HCC had larger tumors (5.9 vs. 4.7 cm, p<0.001), underwent liver resection at a higher rate (66.0% vs. 17.4%, p<0.001), and had better overall survival than the cirrhotic HCC patients (median 5.67 vs. 2.83 years, p<0.001).

### Conclusions

Nearly 26% of the HCCs occurred in patients without cirrhosis.

**Funding:** This study was supported by Grant CMRPG8J1281 from the Kaohsiung Chang Gung Memorial Hospital, Taiwan. Grant Recipient is Yi-Hao Yen. The funders had no role in study design, data collection and analysis, decision to publish, or preparation of the manuscript. There was no additional external funding received for this study.

**Competing interests:** The authors certify that they have no affiliations with or involvement in any organization or entity with any financial interest (such as honoraria; educational grants; participation in speakers' bureaus; membership, employment, consultancies, stock ownership, or other equity interest; and expert testimony or patent-licensing arrangements), or non-financial interest (such as personal or professional relationships, affiliations, and knowledge or beliefs) in the subject matter or materials discussed in this manuscript. This does not alter our adherence to PLOS ONE policies on sharing data and materials.

HBV was the most common liver disease in these patients, and the survival was better in the non-cirrhotic patients than the cirrhotic patients.

## Introduction

Hepatocellular carcinoma (HCC) is one of the most common causes of cancer death worldwide [1,2]. Non-alcoholic fatty liver disease (NAFLD) is the most prevalent chronic liver disease worldwide. The prevalence of NAFLD in Taiwan is 33.3% [3]. NAFLD is one of the leading etiologies of HCC in Western countries [4]. In contrast, a previous study reported that the prevalence of NAFLD-associated HCC is only 5% in Taiwan [5].

In Western countries, HCC arises in a cirrhotic liver in up to 90% of cases [6].

Accordingly, the current American Association for the Study of Liver Diseases (AASLD) guidelines recommend surveillance for HCC in those with cirrhosis of any etiology. The only non-cirrhotic patients requiring such surveillance, meanwhile, are selected hepatitis B virus (HBV) patients (e.g. Asian male HBV carriers over age 40, Asian female HBV carriers over age 50, HBV carrier with family history of HCC and African and/or North American blacks with HBV) [4]. However, in a relevant proportion of patients with NAFLD, HCC develops in non-cirrhotic livers [7,8]. At present, there is insufficient evidence to modify the current guideline definitions of patients at risk for HCC. However, grey areas do exist. In particular, an exact estimate of the risk of HCC development in non-cirrhotic patients with NAFLD remains unavailable. A recent study from the United States (US) reported, however, that nearly 12% of HCCs occurred in patients without cirrhosis, with NAFLD being the most common liver disease in these patients [9].

It is unclear whether the rising incidence of NAFLD globally has resulted in higher numbers of non-cirrhotic NAFLD-associated HCCs, especially in HBV endemic areas where the leading etiology of non-cirrhotic HCC is HBV.

To illuminate these issues, we studied 2055 patients with de novo HCC who were managed at an academic medical center in East Asia from 2011 through 2017. We aimed to evaluate (1) the characteristics and frequency of non-cirrhotic HCC, (2) the contribution of NAFLD-associated HCC to the burden of non-cirrhotic HCC, and (3) the treatment modalities applied to non-cirrhotic HCC patients and their survival compared with cirrhotic patients with HCC.

## Patients and methods

### Patients

Data were extracted from the Kaohsiung Chang Gung Memorial Hospital HCC registry database. We consecutively enrolled de novo HCC patients managed at our institution from January 2011 to December 2017. A flow chart of the patient enrollment is shown in Fig 1. Ultimately, a total of 2055 patients were enrolled in this study. The data in the Kaohsiung Chang Gung Memorial Hospital HCC registry database were updated every 1 years. The personnel who register the cancer registry data can also check the vital statuses of patients with cancer by using a website maintained by the Ministry of Health and Welfare, Taiwan (https://hosplab.hpa.gov.tw/CSTIIS/index.aspx). The diagnosis of HCC was based on the recommendations of international guidelines [7–11] and/or multidisciplinary team discussions. The presence of cirrhosis (as indicated by Ishak fibrosis stage 5 or 6) [10] was assessed, whenever possible, by histology; in the remaining cases, it was assessed by image studies. The imaging features of cirrhosis are as follows: nodular liver surface, blunt edge, small liver size with

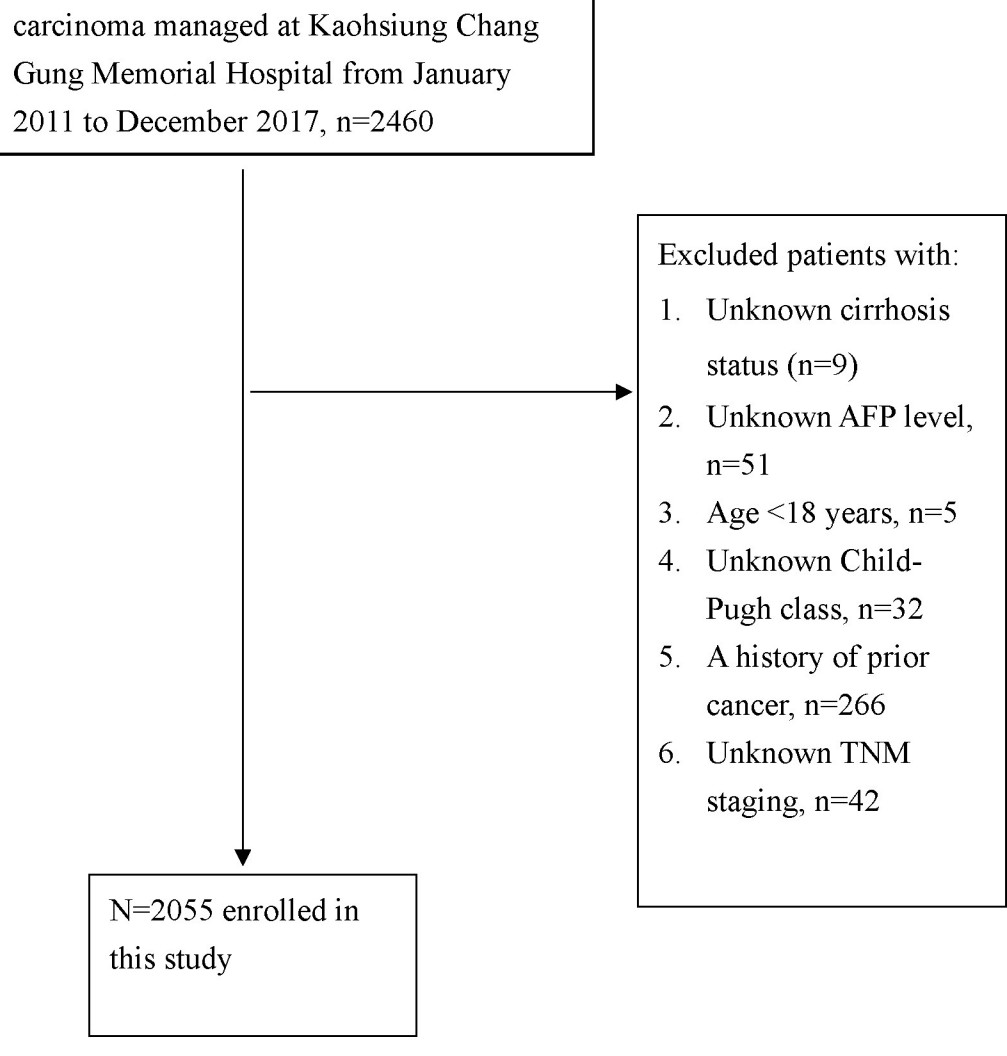

**Fig 1. Flow chart of the patients' enrollment.**

relative enlargement of the caudate lobe or left lobe, coarse or heterogeneous liver parenchyma, and regenerative nodules [12]. The 7[th] version of the American Joint Committee on Cancer (AJCC) tumor-node-metastasis (TNM) staging was applied in this study [11]. Alcohol use disorder was defined as habitual drinking.

In line with the study conducted by Gawrieh et al. [9], having hepatitis C virus (HCV) (which was indicated by testing anti-HCV positive) was considered the primary etiology of liver disease regardless of other potential etiologies. In other words, the etiology for a given patient could be regarded as HCV and hepatitis B virus (HBV), HCV and alcohol, or HCV alone, whereas having HBV (which was indicated by testing hepatitis B surface antigen (HBsAg) positive) was only considered the primary etiology of liver disease if no HCV was present. In such cases, the etiology for a given patient could be regarded as HBV and alcohol or HBV alone.

This study was approved by the Institutional Review Board of Kaohsiung Chang Gung Memorial Hospital (reference number: 202000398B0). The IRB waived the need for informed consent.

Information about the patient records used in our retrospective study: a) all data were not anonymized before we accessed them; b) the date range during which patients' medical records were accessed was January 2011- September 2019.

## Statistical analysis

The baseline characteristics and treatment modalities of the patients with and without cirrhosis were summarized as frequencies (percentages) or means (standard deviations). For categorical variables, the differences between the groups were estimated by the chi-squared test or Fisher's exact test. For continuous variables, the differences between the groups were estimated by the independent two sample t-test. The median overall survival (OS) of both groups was reported with 95% confidence intervals (CIs). The OS curves of the cirrhotic group and the non-cirrhotic group were illustrated using Kaplan-Meier estimation and tested using the log-rank test. Cox proportional hazard regression was used to evaluate the variables associated with mortality among the patients without cirrhosis. The covariates in the multivariate model were chosen a priori for clinical importance. Potential confounders included age, sex, Child-Pugh class, TNM stage, alpha-fetoprotein (AFP) level, and treatment. These variables were always retained in the multivariate regression analysis. *P* values less than 0.05 were considered statistically significant. All analyses were performed using the Stata version 14.0 software (StataCorp, 2015, Stata Statistical Software: Release 14. College Station, TX: StataCorp LP).

## Results

### Characteristics of the patients with HCC with and without cirrhosis

Among the 2055 patients enrolled in this study, 529 (25.7%) did not have cirrhosis.

Histology was available in 670 patients (32.6%). Among 670 patients, 87 patients had an Ishak fibrosis score of 5, and 234 patients had an Ishak fibrosis score of 6. The remaining 349 patients were non-cirrhotic. Among the 349 non-cirrhotic patients, 169 (48.4%) patients had lower stages of fibrosis (i.e. Ishak fibrosis scores of 0–2, which represent the absence of bridging fibrosis), and 180 (51.6%) patients had higher stages of fibrosis (i.e. Ishak fibrosis scores of 3–4, which represent the presence of bridging fibrosis) [10].

Compared to those with cirrhosis, the patients without cirrhosis were younger, included a greater proportion of males, and had a lower mean body mass index (BMI). As expected, the non-cirrhotic patients included a higher proportion of patients with Child-Pugh class A liver disease, had lower bilirubin levels, and had lower international normalized ratio (INR) values than the cirrhotic patients. With regard to the underlying liver disease, HBV was more common in the non-cirrhotic patients than the cirrhotic patients (Table 1).

### Tumor characteristics in patients with and without cirrhosis

Compared with the cirrhotic patients, the non-cirrhotic patients had larger tumors, included a higher proportion of patients with a pathological diagnosis of HCC, included a higher proportion with AJCC TNM stage 1, included a lower proportion with AJCC TNM stage 2, included a higher proportion with moderate tumor differentiation, included a higher proportion with Barcelona Clinic Liver Cancer (BCLC) stage A or B, and included a lower proportion with BCLC 0, C, or D (Table 2).

### Treatment and survival of patients with non-cirrhotic HCC

Compared with the cirrhotic patients, a higher proportion of the non-cirrhotic patients received liver resection and a lower proportion of the non-cirrhotic patients received

**Table 1. Characteristics of the patients with hepatocellular carcinoma with and without underlying cirrhosis.**

| Variables | Cirrhosis no, N = 529 | Cirrhosis yes, N = 1526 | P |
|---|---|---|---|
| Age (year) | 60.9 ± 12.5 | 62.5 ± 11.2 | 0.006 |
| Male | 413 (78.1%) | 1088 (71.3%) | 0.002 |
| BMI(kg/m$^2$) | 24.3 ± 3.9 | 25.3 ± 4.1 | <0.001 |
| Child Pugh classification | | | <0.001 |
| A | 492 (93%) | 1143 (74.9%) | |
| B or C | 37 (7.0%) | 383 (25.1%) | |
| Creatinine (mg/dL) | 1.61 ± 2.31 | 1.49 ± 1.93 | 0.24 |
| Bilirubin (mg/dL) | 1.42 ± 2.94 | 2.31 ± 4.04 | <0.001 |
| INR | 1.05 ± 0.26 | 1.16 ± 0.39 | <0.001 |
| Liver etiology | | | <0.001 |
| HBV | 259 (49.0%) | 641 (42.0%) | |
| HCV | 162 (30.6%) | 619 (40.6%) | |
| Alcohol use disorder | 14 (2.6%) | 57 (3.7%) | |
| All negative | 94 (17.8%) | 209 (13.7%) | |

BMI, body mass index; INR, International Normalized Ratio; HBV, hepatitis B virus; HCV, hepatitis C virus.

radiofrequency ablation or transcatheter arterial embolization/ transcatheter arterial chemoembolization (Table 3). The 5-year OS rate of the non-cirrhotic patients was 59.5%, while that of the cirrhotic patients was 37.1% (p<0.001, Fig 2). The median survival duration was 5.67 (95% CI = 5.42-not available) years in the non-cirrhotic patients and 2.83 (95% CI = 2.5–3.17) years in the cirrhotic patients (p<0.001) (Note: "not available" indicates that the 95% CI had not been reached yet). Multivariate analysis showed that age > 70 years, AJCC TNM stages 3 and 4, serum AFP level > 200 ng/ml, and non-curative treatments were associated with mortality (Table 4).

## Diagnostic accuracy of imaging for cirrhosis

We randomly selected 202 patients who had undergone surgical resection to evaluate the correlation between imaging findings and histology for cirrhosis. We reviewed the computed tomography (CT) reports or magnetic resonance imaging (MRI) reports (if the former is not available) of these patients. Among 202 patients, 20 patients had alcohol use disorder, 112 patients were HBsAg positive, 52 patients were anti-HCV positive, 131 patients were diagnosed as non-cirrhotic by imaging (109 patients were histologically non-cirrhotic on histology, 22 patients were histologically cirrhotic), 71 patients were diagnosed as cirrhotic by imaging (30 patients were histologically non-cirrhotic, 41 patients were histologically cirrhotic). Using histology as a reference, the imaging diagnostic accuracy, sensitivity, specificity, positive predictive value, and negative predictive value for cirrhosis were 74.3%, 65.1%, 78.4%, 57.7%, and 83.2%, respectively.

## Discussion

In the present study, nearly 26% of the patients were non-cirrhotic. The non-cirrhotic patients were younger, included a higher proportion of male patients, and had a lower mean BMI than the cirrhotic patients. This phenomenon could be explained by HBV being the leading etiology among the non-cirrhotic patients. HBV-associated HCC is seen in younger and predominately male patients compared with HCCs with other etiologies [4,13–15]. Relatedly, obesity is a risk factor for fibrosis progression, not only in NAFLD, but also in other chronic liver diseases [16].

**Table 2. Tumor characteristics in patients with and without underlying cirrhosis.**

| Variables | Cirrhosis no, N = 529 | Cirrhosis yes, N = 1526 | P |
|---|---|---|---|
| Tumor size (cm) | 5.9 ± 4.9 | 4.7 ± 4.1 | <0.001 |
| AFP (ng/ml) | | | <0.001 |
| <20 | 289 (54.6%) | 712 (46.7%) | |
| 20–200 | 90 (17.0%) | 378 (24.8%) | |
| >200 | 150 (28.4%) | 436 (28.6%) | |
| HCC diagnosis | | | <0.001 |
| pathology | 464 (87.7%) | 798 (52.3%) | |
| Clinical | 65 (12.3%) | 728 (47.7%) | |
| Image 7th AJCC TNM | | | <0.001 |
| 1 | 336 (63.5%) | 699 (45.8%) | |
| 2 | 48 (9.1%) | 375 (24.6%) | |
| 3A | 38 (7.2%) | 100 (6.6%) | |
| 3B | 33 (6.2%) | 160 (10.5%) | |
| 3C | 35 (6.6%) | 54 (3.5%) | |
| 4A | 3 (0.6%) | 22 (1.4%) | |
| 4B | 36 (6.8%) | 116 (7.6%) | |
| Image 7th AJCC TNM | | | 0.34 |
| 1 or 2 | 384 (72.6%) | 1074 (70.4%) | |
| 3 or 4 | 145 (27.4%) | 452 (29.6%) | |
| Tumor differentiation | | | 0.003 |
| Well | 46 (8.7%) | 128 (8.4%) | |
| Moderate | 367 (69.4%) | 552 (36.2%) | |
| Poor | 20 (3.8%) | 40 (2.6%) | |
| Undifferentiated | 1 (0.2%) | 1 (0.1%) | |
| BCLC stage | | | <0.001 |
| 0 | 64 (12.1%) | 238 (15.6%) | |
| A | 221 (41.8%) | 561 (36.8%) | |
| B | 129 (24.4%) | 247 (16.2%) | |
| C | 97 (18.3%) | 363 (23.8%) | |
| D | 18 (3.4%) | 117 (7.7%) | |

7th The American Joint Committee on Cancer / Tumor size, Lymph Nodes affected, Metastases (AJCC TNM) was adopted; HCC, hepatocellular carcinoma; AFP, Alpha fetoprotein; BCLC, Barcelona Clinic Liver Cancer.

**Table 3. Treatment modalities offered to patients with HCC, stratified according to cirrhosis status.**

| Variables | Cirrhosis no, N = 529 | Cirrhosis yes, N = 1526 | P |
|---|---|---|---|
| Treatment | | | <0.001 |
| Resection | 349 (66.0%) | 265 (17.4%) | |
| Transplant | 3 (0.6%) | 62 (4.1%) | |
| RFA | 67 (12.7%) | 418 (27.4%) | |
| TACE/TAE | 39 (7.4%) | 465 (30.5%) | |
| Sorafenib | 23 (4.3%) | 94 (6.2%) | |
| BSC | 18 (3.4%) | 125 (8.2%) | |
| Other | 30 (5.7%) | 97 (6.4%) | |

Other treatment (i.e. systemic chemotherapy, hepatic artery infusion chemotherapy or external beam radiation therapy). RFA, radiofrequency ablation; TACE/TAE, transcatheter arterial chemoembolization/embolization; BSC, best supportive care.

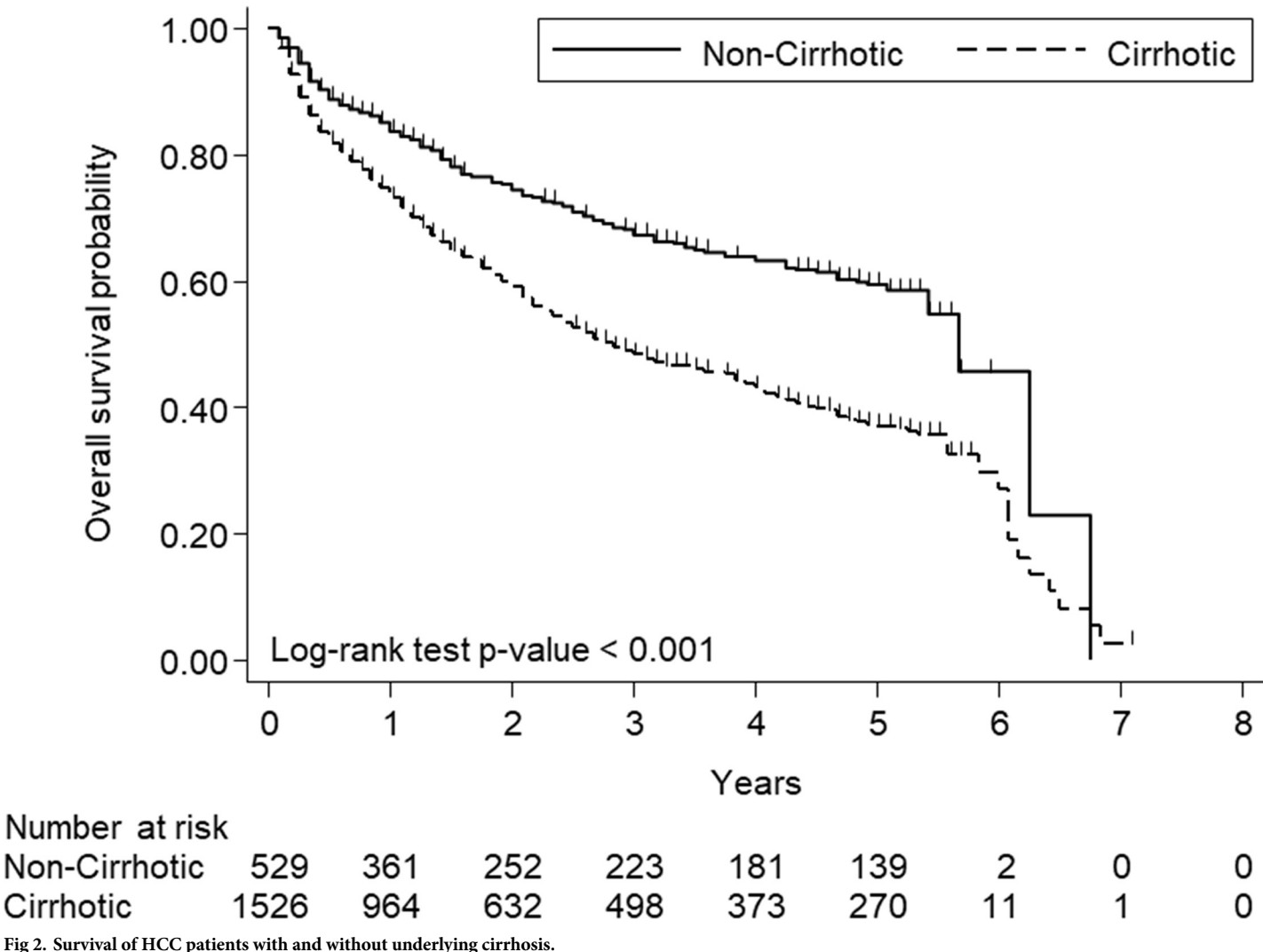

**Fig 2. Survival of HCC patients with and without underlying cirrhosis.**

The leading etiology of HCC among the non-cirrhotic patients was HBV, as expected, followed by HCV. HBV infection is a risk factor for non-cirrhotic HCC in East Asia [17]. Notably, 162 (20.7%) of the patients who tested anti-HCV positive in the present study were non-cirrhotic. We did not have information regarding whether these patients had viremia. HCV-associated HCC usually arises from cirrhotic liver [4,13–15], and the finding that a significant proportion of the HCV patients in the present study were non-cirrhotic could have been due to the study having included patients with past resolved HCV infections and patients who received antiviral therapy with sustained virological responses which led to cirrhosis regression [18].

The overall rate of non-cirrhotic HCC in this study (25.7%) was higher than that in the previous study from the US (11.7%) [9]. This discrepancy between the two studies could have been due to the different definitions of cirrhosis used and referral bias. There are no universally accepted criteria for defining cirrhosis in the absence of histology. Previous studies have used images, laboratory results, and the presence or absence of liver-related complications to define cirrhosis in those without histological evidence. In contrast, we used imaging alone to

**Table 4. Variables associated with mortality among patients with non-cirrhotic HCC (multivariate regression analysis).**

| Variables | Multivariate | P value |
|---|---|---|
| | HR (95% CI) | |
| Age (years) | | |
| ≤ 70 | 1.00 | |
| > 70 | 1.69 (1.20–2.39) | 0.003 |
| Sex | | |
| Female | 1.00 | |
| Male | 1.24 (0.83–1.85) | 0.29 |
| Child Pugh class | | |
| A | 1.00 | |
| B or C | 1.63 (0.96–2.76) | 0.07 |
| Image 7th AJCC TNM | | |
| 1 | 1.00 | |
| 2 | 0.84 (0.41–1.73) | 0.63 |
| 3 | 2.85 (1.89–4.29) | <0.001 |
| 4 | 10.22 (5.52–18.92) | <0.001 |
| AFP (ng/ml) | | |
| ≤ 200 | 1.00 | |
| > 200 | 1.55 (1.11–2.19) | 0.01 |
| Treatment | | |
| Curative treatments | 1.00 | |
| TAE/TACE | 3.01 (1.85–4.90) | <0.001 |
| Sorafenib | 2.67 (1.46–4.87) | 0.001 |
| BSC | 15.31 (7.61–30.81) | <0.001 |
| others | 3.46 (2.00–5.97) | <0.001 |

Curative treatment (i.e. transplant, resection or radiofrequency ablation); Other treatment (i.e. systemic chemotherapy, hepatic artery infusion chemotherapy or external beam radiation therapy). TACE/TAE, transcatheter arterial chemoembolization/embolization; BSC, best supportive care.

define cirrhosis in those without histological evidence. The aforementioned previous study from the US was a multi-center study, and referral is needed in the US. As such, there was referral bias in the previous study [9]. In contrast, referral is not needed in Taiwan.

In the present study, the average tumor size was significantly larger in the non-cirrhotic patients than in the cirrhotic patients, possibly due to a lack of surveillance in the non-cirrhotic patients. 66.0% of the non-cirrhotic patients underwent liver resection. In contrast, only 17.4% of the patients with cirrhosis underwent liver resection. Major liver resections can be performed with low rates of life-threatening complications in non-cirrhotic patients [13]. The better survival in non-cirrhotic patients may be due to liver resection being associated with improved survival across BCLC stages [19]. Furthermore, cirrhosis is a major risk factor for HCC recurrence, and the relatively decreased incidence of recurrence in those without cirrhosis also leads to better survival. Finally, non-cirrhotic patients are less likely to develop liver decompensation after multiple liver-directed therapies than cirrhotic patients, and cirrhosis is the most important competing risk of death in patients with HCC [4,13].

Old age, advanced tumor stage, and non-curative treatments were associated with mortality in the non-cirrhotic patients. The results of the present study are consistent in this regard with those of previous studies [4,13–15].

Our HCC registry database only records etiologies including HBV, HCV, and alcohol use disorder; other etiologies, including NAFLD, are not recorded. Current guidelines recommend that NAFLD be diagnosed based on evidence of steatosis, whether provided by histology or imaging, in the absence of an alternative liver disease [16].

The gold standard for diagnosing steatosis is liver biopsy or magnetic resonance spectroscopy. However, liver biopsy is invasive and magnetic resonance spectroscopy is expensive. Therefore, ultrasound is the most commonly used imaging modality for diagnosing steatosis [3]. Ultrasound has good sensitivity (85%) and specificity (94%) for diagnosing moderate to severe steatosis [20], but it is less reliable for diagnosing mild steatosis. Furthermore, over time, steatosis may disappear when cirrhosis develops (i.e. burnout non-alcoholic steatohepatitis) [16].

Thus, previous studies used different criteria to define NAFLD-associated HCC in the absence of histology. In the absence of an alternative liver disease as a precondition, Mittal et al. defined NAFLD as the presence of metabolic syndrome [21]. Bengtsson et al. defined NAFLD as BMI $\geq$25 kg/m$^2$ and Type 2 diabetes, or BMI $\geq$30 kg/m$^2$ [22]. Kanwal et al. defined NAFLD as elevated alanine aminotransferase values [23]. Younossi et al. defined NAFLD by using the International Classification of Diseases codes, and including the codes for NAFLD and cryptogenic liver disease [24]. A lifestyle history, including factors such as a sedentary lifestyle, eating habits, and the trajectories of weight change since young adulthood and waist expansion, is a prerequisite for diagnosing NAFLD [25].

The limited number of patients with alcohol use disorder (n = 71) in the present study suggests that the reported alcohol use of the patients may have been underestimated. Thus, the patients with apparently non-viral- and non-alcohol-related etiologies in the present study may actually have been composed in part of some with etiologies including NAFLD, unreported alcohol use disorder, rare chronic liver diseases, and HBV with HBsAg seroclearance [26] in an HBV endemic area, namely Taiwan. Among the entire cohort, only 94 patients (4.6%) in the present study were found to have non-cirrhotic, non-viral-, non-alcohol-related HCC, which indicated that the number of patients with NAFLD-associated non-cirrhotic HCC should have been less than 94.

Our previous study enrolled 5613 consecutive patients with HCC who were treated at our institution between 1986 and 2002, including 4287 (76.4%) patients with HCC associated with viral hepatitis [27]. In the present study, 1681 (81.8%) patients had HCC associated with viral hepatitis. As time goes by, the proportion of patients with non-viral-associated HCC has not significantly changed. According to a recent review article, although the potential impact of NAFLD on HCC incidence in North America is major, the potential impact of NAFLD in the epidemiology of HCC in Asia is minor at present but may be growing [28].

In Taiwan, surveillance for HCC in those with chronic viral hepatitis and cirrhosis of any etiology is reimbursed by the National Health Insurance program. Therefore, those patients with non-viral, non-cirrhotic HCC may be diagnosed incidentally or during a symptom work-up. It is impossible to surveil (or even screen) for NAFLD patients without cirrhosis. NAFLD-associated HCC in non-cirrhotic liver was therefore likely underestimated in the present study and in other studies [9,21–24].

One strength of the present study is that it had no referral bias. The patients enrolled in the present study could thus be representative of the general HCC population in Taiwan. Our institution is one of the largest academic medical cancer treatment centers in Taiwan, and has 2724 beds and 861 physicians (https://en.wikipedia.org/wiki/Chang_Gung_Medical_Foundation#Kaohsiung_Chang_Gung_Memorial_Hospital). The current healthcare system in Taiwan, known as the National Health Insurance program, was instituted in 1995, and the population coverage had reached 99% as of 2004. Under the system, citizens can choose

hospitals and physicians without referral, and regular office visits have co-payments as low as US$5 per visit. Therefore, most patients are quick to visit medical centers if they feel some need to. The main island of Taiwan, measuring 35808 square kilometers, making it smaller in size to Switzerland, is a highly urbanized island with 26 academic medical centers, all located on west side of the island (https://en.wikipedia.org/wiki/Healthcare_in_Taiwan). Thus, there is no barrier to healthcare for citizens who live on the west side of the main island. A second strength of the present study is that we used an authoritative source to check vital statuses of the patients enrolled. We could thus make sure of the vital status of every single patient enrolled in the study.

There were also some limitations in the present study. First, this was a retrospective study. Second, there are no specific diagnostic criteria for defining the presence vs. absence of cirrhosis on imaging in our clinical practice. Relatedly, the diagnosis of cirrhosis on imaging was subjective in the present study. Third, there was a lack of etiology data on patients without HBV, HCV, or alcohol-related liver disease (such as NAFLD). Diabetes, hypertension, dyslipidemia, and other risk factors typically associated with NAFLD (e.g. metabolic syndrome, truncal obesity defined by waist circumference) were not recorded in our HCC registry data. However, the HCC registry data of our institution do include BMI data. Obesity is defined as BMI $\geq 27 (kg/m^2)$ by the Health Promotion Administration, Ministry of Health and Welfare, Taiwan (https://health99.hpa.gov.tw/onlinkhealth/onlink_bmi.aspx). Among those with HBV, HCV, or alcohol-related liver disease, there were 443 (25.29%) patients with BMI $\geq 27 (kg/m^2)$. Among those with unknown etiology of liver disease, there were 95 (31.35%) patients with BMI $\geq 27 (kg/m^2)$. The proportion of patients with BMI $\geq 27 (kg/m^2)$ was, therefore, significantly higher among those with unknown etiology of liver disease compared with those with HBV, HCV, or alcohol-related liver disease (p = 0.002).

The limitations of our HCC registry data were as follows: (1) only the first-line therapy was recorded, (2) the patients' reported alcohol consumption levels may have been underestimated. The HCC registry data for our institution do not, however, include daily alcohol intake or data provided by the use of screening tools [e.g. the AUDIT (Alcohol Use Disorders Inventory Test)] to identify alcohol use disorders [29] and (3) data regarding medical comorbidities, abstinence from alcohol, antiviral therapy, and weight change during the follow-up period, which might affect the prognosis, were not available [30–33].

In conclusion, around 26% of the patients with HCC in our large cohort of patients from an academic medical center in East Asia were non-cirrhotic. HBV and HCV were the leading etiologies of the investigated patients with HCC, regardless of whether they did or did not have cirrhosis. Although obesity rates are rising worldwide [34], the contribution of NAFLD-associated non-cirrhotic HCC to the overall burden of HCC was not significant enough to result in an increase in the total burden of HCC in Taiwan as of the end of this study in 2017.

## Supporting information

**S1 File. Raw data of this cohort.**
(XLSX)

## Acknowledgments

The authors thank Cancer Center, Kaohsiung Chang Gung Memorial Hospital for the provision of HCC registry data. The authors thank Chih-Yun Lin and Nien-Tzu Hsu and the Biostatistics Center, Kaohsiung Chang Gung Memorial Hospital for statistics work.

## Author Contributions

**Conceptualization:** Yi-Hao Yen.

**Supervision:** Yu-Fan Cheng, Jing-Houng Wang, Chih-Che Lin, Chih-Chi Wang.

**Writing – original draft:** Yi-Hao Yen.

**Writing – review & editing:** Yi-Hao Yen.

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
