## [Decision Letter · Decision Letter 0]

19 Oct 2020

PONE-D-20-27109

Characteristics and etiologies of hepatocellular carcinoma in patients without cirrhosis: when East meets West

PLOS ONE

Dear Dr. Yen,

Thank you for submitting your manuscript to PLOS ONE. After careful consideration, we feel that it has merit but does not fully meet PLOS ONE’s publication criteria as it currently stands. Therefore, we invite you to submit a revised version of the manuscript that addresses the points raised during the review process. In particular concerning patients description and result's analysis.

We look forward to receiving your revised manuscript.

Kind regards,

Isabelle Chemin, PhD

Academic Editor

PLOS ONE

Journal Requirements:

2. In the ethics statement in the manuscript and in the online submission form, please provide additional information about the patient records used in your retrospective study, including: a) whether all data were fully anonymized before you accessed them; and b) the date range (month and year) during which patients' medical records were accessed.

 "This study was supported by Grant CMRPG8J1281 from the Chang Gung Memorial Hospital-Kaohsiung Medical Center, Taiwan. No conflict of interests.

i) Please provide an amended statement that declares *all* the funding or sources of support (whether external or internal to your organization) received during this study, as detailed online in our guide for authors at http://journals.plos.org/plosone/s/submit-now.  Please also include the statement “There was no additional external funding received for this study.” in your updated Funding Statement.

ii) Please include your amended Funding Statement within your cover letter. We will change the online submission form on your behalf.

Reviewers' comments:

Reviewer's Responses to Questions

**Comments to the Author**

1. Is the manuscript technically sound, and do the data support the conclusions?

Reviewer #1: Partly

2. Has the statistical analysis been performed appropriately and rigorously? 

Reviewer #1: Yes

3. Have the authors made all data underlying the findings in their manuscript fully available?

Reviewer #1: Yes

4. Is the manuscript presented in an intelligible fashion and written in standard English?

Reviewer #1: Yes

5. Review Comments to the Author

Reviewer #1: Characteristics and etiologies of hepatocellular carcinoma in patients without cirrhosis: when East meets West

Yen et al.

Yen et al. have performed a retrospective review of patients in their center’s HCC registry database and identified patients who presented with HCC in the presence vs. absence of cirrhosis, with the goal of identifying patient characteristics and outcomes associated with non-cirrhotic HCC. This study includes a large sample (529 total patients) who presented with HCC in the absence of cirrhosis. There are certain elements of the manuscript that require further elaboration and clarification. The strengths of the cohort include accurate treatment and survival data given the Taiwanese healthcare system.

Major issues/comments:

1) This reviewer has several comments/questions related to the criteria for “no cirrhosis”, specifically:

a. Please provide the specific diagnostic criteria for defining the presence vs. absence of cirrhosis on imaging. For the patients who had histologic diagnosis available, did the findings on histology correlate with the findings on imaging? How was evidence of portal hypertension adjudicated?

b. In the patients who had the presence or absence of cirrhosis assessed by imaging (but not histology), how many had ultrasound vs. CT scan vs. MRI? In the discussion section, the authors comment on the sensitivity and specificity of ultrasound in diagnosing cirrhosis, which leads this reviewer to believe that most patients in this cohort who had an imaging diagnosis of cirrhosis vs. no cirrhosis had this diagnosis made by ultrasound. To perform TNM staging for patients with HCC, evaluation for metastatic disease typically requires cross-sectional imaging such as CT or MRI. If the majority of patients therefore had cross-sectional imaging (and not only ultrasound), this information about ultrasound specificity and sensitivity for cirrhosis in the authors’ previous study does not seem relevant.

2) The second major concern of this reviewer is the lack of data on patients without HBV, HCV, or alcohol-related liver disease. This reviewer believes that this is a major limitation and warrants acknowledgement, as 17.8% of patients have an unknown underlying etiology of liver disease. The statement that only 4.6% of patients in the entire cohort might have NAFLD-associated, non-cirrhotic HCC minimizes this limitation.

a. This reviewer asks the authors to consider including this as a study limitation.

b. Are specific patient characteristics known for those with unknown etiology of liver disease? I.e. do these patients have increased rates of diabetes, hypertension, dyslipidemia, obesity or other risk factors we would typically associate with NAFLD?

Minor issues/comments:

1) For those patients with histologic diagnosis of no cirrhosis – is there more detailed information on the stage of fibrosis other than “no cirrhosis”? Are patients without cirrhosis but higher stages of fibrosis, for example, more at risk of developing non-cirrhotic HCC than the patients without cirrhosis and lower stages of fibrosis? Were any noninvasive measurements (FIB-4, NAFLD-FS, APRI) calculated to help estimate numbers?

2) Do the authors have information on patient medical comorbidities? Are there certain comorbidities with associated increased rates of non-cirrhotic HCC?

3) Please define “habitual drinking” by standards associated with diagnostic criteria.

6. PLOS authors have the option to publish the peer review history of their article (what does this mean?). If published, this will include your full peer review and any attached files.

Reviewer #1: No

---

## [Author Response · Author response to Decision Letter 0]

13 Nov 2020

Reviewer #1: Characteristics and etiologies of hepatocellular carcinoma in patients without cirrhosis: when East meets West

Yen et al.

Yen et al. have performed a retrospective review of patients in their center’s HCC registry database and identified patients who presented with HCC in the presence vs. absence of cirrhosis, with the goal of identifying patient characteristics and outcomes associated with non-cirrhotic HCC. This study includes a large sample (529 total patients) who presented with HCC in the absence of cirrhosis. There are certain elements of the manuscript that require further elaboration and clarification. The strengths of the cohort include accurate treatment and survival data given the Taiwanese healthcare system.

Major issues/comments:

1) This reviewer has several comments/questions related to the criteria for “no cirrhosis”, specifically:

a. Please provide the specific diagnostic criteria for defining the presence vs. absence of cirrhosis on imaging. For the patients who had histologic diagnosis available, did the findings on histology correlate with the findings on imaging? How was evidence of portal hypertension adjudicated?

Response: Thank you so much for your comments. 

1. The imaging features of cirrhosis are as follows: nodular liver surface, blunt edge, small liver size with relative enlargement of the caudate lobe or left lobe, coarse or heterogeneous liver parenchyma, and regenerative nodules [12]. Please see page 7, red color paragraph. In our clinical practice, however, there are no specific diagnostic criteria for defining the presence vs. absence of cirrhosis on imaging. Relatedly, the diagnosis of cirrhosis on imaging was subjective in the present study. please see page 26, line 2-4.

2. Diagnostic accuracy of imaging for cirrhosis:

We randomly selected 202 patients who had undergone surgical resection to evaluate the correlation between imaging findings and histology for cirrhosis. We reviewed the CT reports or MRI reports (if the former is not available) of these patients. Among 202 patients, 20 patients had alcohol use disorder, 112 patients were HBsAg positive, 52 patients were anti-HCV positive, 131 patients were diagnosed as non-cirrhotic by imaging (109 patients were histologically non-cirrhotic on histology, 22 patients were histologically cirrhotic), 71 patients were diagnosed as cirrhotic by imaging (30 patients were histologically non-cirrhotic, 41 patients were histologically cirrhotic). Using histology as a reference, the imaging diagnostic accuracy, sensitivity, specificity, positive predictive value, and negative predictive value for cirrhosis were 74.3%, 65.1%, 78.4 %, 57.7 %, and 83.2 %, respectively. Please see page 19, first paragraph

3. Signs of portal hypertension on imaging studies such as ascites, splenomegaly, and portosystemic collateral vessels (doi: 10.1007/s12072-016-9760-3) were not recorded in our HCC registry data. In our clinical practice, signs of portal hypertension on imaging studies increase the confidence in the diagnosis of cirrhosis. 

b. In the patients who had the presence or absence of cirrhosis assessed by imaging (but not histology), how many had ultrasound vs. CT scan vs. MRI?

Response: Thank you so much for your comments. 

This data was not recorded in our HCC registry data. In our clinical practice, however, ultrasound is performed in all patients with HCC. Contrast-enhanced CT or MRI is performed, meanwhile, in patients with HCC and adequate renal function (i.e. estimated glomerular filtration rate ≥30 ml/min/1.73 m2), whereas non-contrast-enhanced MRI is performed in patients with HCC and severe renal impairment (i.e. estimated glomerular filtration rate <30 ml/min/1.73 m2). 

In the discussion section, the authors comment on the sensitivity and specificity of ultrasound in diagnosing cirrhosis, which leads this reviewer to believe that most patients in this cohort who had an imaging diagnosis of cirrhosis vs. no cirrhosis had this diagnosis made by ultrasound. To perform TNM staging for patients with HCC, evaluation for metastatic disease typically requires cross-sectional imaging such as CT or MRI. If the majority of patients therefore had cross-sectional imaging (and not only ultrasound), this information about ultrasound specificity and sensitivity for cirrhosis in the authors’ previous study does not seem relevant.

Response: Thank you so much for your comments. 

We have deleted the information about ultrasound specificity and sensitivity for cirrhosis from our previous study.

2) The second major concern of this reviewer is the lack of data on patients without HBV, HCV, or alcohol-related liver disease. This reviewer believes that this is a major limitation and warrants acknowledgement, as 17.8% of patients have an unknown underlying etiology of liver disease. The statement that only 4.6% of patients in the entire cohort might have NAFLD-associated, non-cirrhotic HCC minimizes this limitation.

a. This reviewer asks the authors to consider including this as a study limitation.

b. Are specific patient characteristics known for those with unknown etiology of liver disease? I.e. do these patients have increased rates of diabetes, hypertension, dyslipidemia, obesity or other risk factors we would typically associate with NAFLD?

Response: Thank you so much for your comments. 

1. We have now mentioned the unknown underlying etiology of liver disease in patients without HBV, HCV, or alcohol-related liver disease as a study limitation, as follows: “There were also some limitations in the present study … Third, there was a lack of etiology data on patients without HBV, HCV, or alcohol-related liver disease (such as NAFLD).”please see page 26

2. Diabetes, hypertension, dyslipidemia, and other risk factors typically associated with NAFLD (e.g. metabolic syndrome, truncal obesity defined by waist circumference) were not recorded in our HCC registry data. However, the HCC registry data of our institution do include BMI data. Obesity is defined as BMI ≥27(kg/m2) by the Health Promotion Administration, Ministry of Health and Welfare, Taiwan (https://health99.hpa.gov.tw/onlinkhealth/onlink_bmi.aspx). Among those with HBV, HCV, or alcohol-related liver disease, there were 443 (25.29%) patients with BMI ≥27(kg/m2). Among those with unknown etiology of liver disease, there were 95 (31.35%) patients with BMI ≥27(kg/m2). The proportion of patients with BMI ≥27 (kg/m2) was, therefore, significantly higher among those with unknown etiology of liver disease compared with those with HBV, HCV, or alcohol-related liver disease (p=0.002). please see page 26

Minor issues/comments:

1) For those patients with histologic diagnosis of no cirrhosis – is there more detailed information on the stage of fibrosis other than “no cirrhosis”? Are patients without cirrhosis but higher stages of fibrosis, for example, more at risk of developing non-cirrhotic HCC than the patients without cirrhosis and lower stages of fibrosis? Were any noninvasive measurements (FIB-4, NAFLD-FS, APRI) calculated to help estimate numbers?

Response: Thank you so much for your comments. 

1. Among 670 patients, 87 patients had an Ishak fibrosis score of 5, and 234 patients had an Ishak fibrosis score of 6. The remaining 349 patients were non-cirrhotic. Among the 349 non-cirrhotic patients, 169 (48.4%) patients had lower stages of fibrosis (i.e. Ishak fibrosis scores of 0-2, which represent the absence of bridging fibrosis), and 180 (51.6%) patients had higher stages of fibrosis (i.e. Ishak fibrosis scores of 3-4, which represent the presence of bridging fibrosis) [10]. Please see page 9 and 10, red color paragraph.

2. The HCC registry data for our institution do not include AST, ALT, platelet count, fasting glucose, diabetes, or albumin levels. Relatedly, noninvasive measurements (FIB-4, NAFLD-FS, APRI) were not available in the present study.

2) Do the authors have information on patient medical comorbidities? Are there certain comorbidities with associated increased rates of non-cirrhotic HCC?

Response: Thank you so much for your comments. 

The HCC registry data for our institution do not include patient medical comorbidities. We have including this as a study limitation. Please see page 27, line 5. 

3) Please define “habitual drinking” by standards associated with diagnostic criteria.

Response: Thank you so much for your comments.

Drinking behavior is recorded in our HCC registry data as follows: 

the 000 code refers to someone who has never had an alcoholic drink in their life, while the 001 code refers to someone who may have drank alcohol in the past but is currently abstinent. Code 002 indicates occasional drinking. Code 003 indicates habitual drinking for more than 10 years. Code 004 indicates habitual drinking for less than or equal to 10 years. Code 009 indicates habitual drinking for unknown number of years. The HCC registry data for our institution do not, however, include daily alcohol intake or data provided by the use of screening tools [e.g. the AUDIT (Alcohol Use Disorders Inventory Test)] to identify alcohol use disorders [29]. We have including this as a study limitation. Please see page 27, first paragraph

---

## [Decision Letter · Decision Letter 1]

21 Dec 2020

Characteristics and etiologies of hepatocellular carcinoma in patients without cirrhosis: when East meets West

PONE-D-20-27109R1

Dear Dr. Yen,

We’re pleased to inform you that your manuscript has been judged scientifically suitable for publication and will be formally accepted for publication once it meets all outstanding technical requirements.

Kind regards,

Isabelle Chemin, PhD

Academic Editor

PLOS ONE

Additional Editor Comments (optional):

Reviewers' comments:

Reviewer's Responses to Questions

**Comments to the Author**

1. If the authors have adequately addressed your comments raised in a previous round of review and you feel that this manuscript is now acceptable for publication, you may indicate that here to bypass the “Comments to the Author” section, enter your conflict of interest statement in the “Confidential to Editor” section, and submit your "Accept" recommendation.

Reviewer #1: All comments have been addressed

2. Is the manuscript technically sound, and do the data support the conclusions?

Reviewer #1: Yes

3. Has the statistical analysis been performed appropriately and rigorously? 

Reviewer #1: Yes

4. Have the authors made all data underlying the findings in their manuscript fully available?

Reviewer #1: Yes

5. Is the manuscript presented in an intelligible fashion and written in standard English?

Reviewer #1: Yes

6. Review Comments to the Author

Reviewer #1: Thank you for addressing our questions. The only additional comment to be made is that all limitations with regards to criteria for cirrhosis, absence of classification of portal hypertension be included in the discussion as a limitation as you have done with unknown etiologies of chronic liver disease in your registry, etc.

7. PLOS authors have the option to publish the peer review history of their article (what does this mean?). If published, this will include your full peer review and any attached files.

Reviewer #1: No

---

## [Editor Report · Acceptance letter]

4 Jan 2021

PONE-D-20-27109R1 

Characteristics and etiologies of hepatocellular carcinoma in patients without cirrhosis: when East meets West 

Dear Dr. Yen:

I'm pleased to inform you that your manuscript has been deemed suitable for publication in PLOS ONE. Congratulations! Your manuscript is now with our production department. 

Kind regards, 

on behalf of

Mrs Isabelle Chemin 

Academic Editor

PLOS ONE